# Variable-Length Multiobjective Social Class Optimization for Trust-Aware Data Gathering in Wireless Sensor Networks

**DOI:** 10.3390/s23125526

**Published:** 2023-06-12

**Authors:** Mohammed Ayad Saad, Rosmina Jaafar, Kalaivani Chellappan

**Affiliations:** 1Department of Electrical, Electronics & System Engineering, Faculty of Engineering & Built Environment, Universiti Kebangsaan Malaysia (UKM), Bangi 43600, Selangor, Malaysia; 2Department of Medical Instrumentations Technique Engineering, Al-Kitab University, Kirkuk 36001, Iraq

**Keywords:** variable length, multi-objective, social class optimization, trust aware, data gathering

## Abstract

Data gathering in wireless sensor networks (WSNs) is vital for deploying and enabling WSNs with the Internet of Things (IoTs). In various applications, the network is deployed in a large-scale area, which affects the efficiency of the data collection, and the network is subject to multiple attacks that impact the reliability of the collected data. Hence, data collection should consider trust in sources and routing nodes. This makes trust an additional optimization objective of the data gathering in addition to energy consumption, traveling time, and cost. Joint optimization of the goals requires conducting multiobjective optimization. This article proposes a modified social class multiobjective particle swarm optimization (SC-MOPSO) method. The modified SC-MOPSO method is featured by application-dependent operators named interclass operators. In addition, it includes solution generation, adding and deleting rendezvous points, and moving to the upper and lower class. Considering that SC-MOPSO provides a set of nondominated solutions as a Pareto front, we employed one of the multicriteria decision-making (MCDM) methods, i.e., simple additive sum (SAW), for selecting one of the solutions from the Pareto front. The results show that both SC-MOPSO and SAW are superior in terms of domination. The set coverage of SC-MOPSO is 0.06 dominant over NSGA-II compared with only a mastery of 0.04 of NSGA-II over SC-MOPSO. At the same time, it showed competitive performance with NSGA-III.

## 1. Introduction

Data gathering in wireless sensor networks (WSNs) is a process of collecting data from a field equipped with many sensors and occupying a wide area using single- or multisink WSNs [1,2]. This process is not effective in a large-scale environment due to the increasing cost of deploying a large number of stationary links. Multiple sinks and multiple mobile sinks are more sustainable approaches to managing data transmission to multiple endpoints in a WSN [3]. However, deploying multisink or multiple mobile sinks in a WSN for data gathering is a mathematically complex process. It involves various determinations, namely, the number of mobile sinks [4]; their trajectories, which consist of a sequence of rendezvous points; and selecting [5] the depth or number of hops that are required for collecting the data [6]. The above-stated challenges have encouraged development of a simulator to optimize the performance of wireless sensor networks’ data gathering [7]. This study aimed to compute an optimal decision support algorithm by considering cost, energy consumption, traveling time, and trust [8]. Trust-aware data gathering is regarded as a multiobjective optimization problem because it involves simultaneously optimizing multiple objectives, which are often in conflict with each other. The main objectives of trust-aware data gathering are maximizing the accuracy of the collected data, minimizing the impact of unreliable or malicious sources, maximizing the amount of collected data, and minimizing the cost of data collection. These objectives often conflict with each other, as increasing the amount of collected data may come at the cost of lower accuracy or increased risk of unreliable or malicious sources [9] Correspondingly, improving the accuracy of the data may come at the cost of reducing the amount of data collected. In addition, trust-aware data gathering involves making decisions in a complex and uncertain environment, which makes it challenging to optimize the objectives. The trustworthiness of a source may change over time, or a source may deliberately provide misleading data [10].

Therefore, to address these challenges and optimize conflicting objectives, techniques such as meta-heuristic searching have been extensively used to solve optimization problem with a multiobjective nature [11]. This method starts with the initial setting of random solutions, and it determines the heuristic interactions between the solutions for reaching a set of nondominated solutions after a set of iterations [12]. Multiobjective optimization is evaluated from various perspectives, namely, the Pareto front, hypervolume, delta metric, and generation distance [13]. The quality of the solutions is assessed from various perspectives including domination, diversity, and enabling more choices for the decision-maker [14]. There are numerous algorithms used for various types of meta-heuristics including evolutionary genetic algorithm and particle swarm optimization [15]. However, the meta-heuristic searching algorithm is a specific application that requires developing application-oriented operators that are specifically designed for the problem at hand. These operators are used to generate new candidate solutions during the search process. The effectiveness of these operators can greatly impact the performance of the algorithm and the quality of the solutions obtained [16].

Social class multiobjective particle swarm optimization (SC-MOPSO) is a type of multiobjective particle swarm optimization algorithm that was developed to support variable-length solution spaces [17]. However, SC-MOPSO was developed based on blind mobility operators that do not consider the specific nature of the data-gathering application [18]. Additionally, it has not been extended to be used for multiobjective data gathering with trust-awareness applications using mobile sinks [19].

This study aimed to modify SC-MOPSO for use in multi-mobile sink data-gathering applications with optimal design support capacity. This adaptation involved the development of application-oriented operators and the extension of the algorithm to a multiobjective space with four objectives: trust, energy consumption, cost, and time of collection. The aim was to create an algorithm that can effectively balance these objectives and optimize the data-gathering process for multiple mobile sink systems. In summary, this article proposes an adaptation of the SC-MOPSO algorithm for use in multiple mobile sink data-gathering applications with trust awareness.

## 2. Literature Survey

This section presents a literature survey. It consists of two subsections, namely, trust-aware data-gathering and multicriteria decision-making (MCDM) models that can be used to optimize data gathering in a network.

### 2.1. Trust-Aware Data Gathering

The literature on trust-aware data collection refers to the process of data collection from WSNs, which includes varying degrees of sensor trust. Considering other data-gathering factors, namely, energy consumption, journey time, and cost of the network, necessitates a nonlinear data-gathering mechanism that prioritizes the trajectory of mobile sinks with greater closeness to the trust nodes.

There are many different algorithms that have been used for trust-aware data collection in the literature. A model for gathering reliable data was presented [6] based on edge computing (EC) and the Internet of Things (IoT) The sensor nodes in this model are assessed over a wide range of attributes to obtain accurately measured trust levels. Additionally, by virtually mapping a node’s trust value, the optimum mobility route with high trust was built for mobile data collection. Furthermore, a portable edge data collector was used to access and gather reliable data from sensors with quantifiable degrees of trust [19]. The potential for sliding into local minima when the total amount of virtual forces reaches zero is a limitation of this approach. The model used also only supports one mobile collector. Additionally, the total trust makes the erroneous assumption that there is a linear relationship between several components. In one study [20], it was recommended to use a probabilistic graphical model to completely evaluate the dependability of the sensor nodes utilizing a mobile edge-computing-based intelligent trust-assessment technique. The suggested approach analyzes data collection and communication behavior to assess the reliability of sensor nodes. The proposed mechanism evaluates trust based on data collection and communication behavior and plans the edge nodes’ movement path to increase the likelihood of direct trust evaluation while minimizing travel distance. However, this method ignores the various dimensions of trust and operates with only one mobile collector. Authors [21] also looked at the use of unmanned aerial vehicles (UAVs) as mobile sinks. The problem was recast as a multiobjective joint optimization problem as a result, with a large number of constraints, leading to a typical K-center problem. Then, a lightweight authority authentication model using certain stages was examined to overcome the major security issues with UAV-aided trustworthy data collecting. This architecture needs reliable sensors and UAVs with a tolerable authentication latency to carry out the data-gathering session. Smart data collecting with cloud services (S-SDC), a secure mobile sensor network-based protocol, was suggested in a study using mobile sensors to enable information dissemination in dynamic networks with the least loss and power consumption [22]. Additionally, the method supports authoritative routing and offers network consistency and availability for the collected data with cloud enterprises. The fitness-based fuzzy C-means (Fit-FCM) algorithm, which gathers nodes’ residual energy and utilizes the energy to pick new CHs while discarding nodes with the lowest energy, was proposed as a UAV-based CH selection (CHS) technique for usage in WSNs [23]. The created Fit-FCM algorithm, which takes into consideration fitness functions including energy, distance, and trust, is used to conduct CHS after simulating UAV-based WSN nodes. An IoT-based trust evaluation was carried out [24] to determine whether the target object can consistently carry out defined actions. The target object’s direct interaction history or other recommenders’ suggested data are both considered in the valid time domain. Two types of trust—active trust and recommended trust—were offered to characterize the trust evaluation process for Internet of Things devices. A temperature-aware trusted routing scheme (TTRS), a multifactor routing strategy based on the remaining energy, the hop count, and the trust value of sensor nodes, was proposed [25] to find the shortest and most reliable route. The scheme includes a hotspot node detection algorithm to detect malicious relay nodes and a route discovery and management mechanism. In a study [26], a belief-based trust evaluation mechanism was developed that uses a Bayesian estimation approach to distinguish hostile nodes from trustworthy nodes and protect against attacks such as bad-mouthing, on-off, and denial of service. In the work of [27], an aggregate signature-based trust routing (ASTR) scheme was developed to ensure safe data collection in WSNs. The scheme uses aggregate signatures to maintain data integrity and employs a detour routing scheme to verify the safe delivery of data to the sink. The trustworthiness of a path is determined by its success rate. 

In [28], a trust-based data gathering approach is proposed that focuses on collecting data in a sensor-driven environment through data aggregation and reconstruction. The concept of multidimensional trust was introduced, incorporating communication, data, resource, and forwarding trust. However, the approach has limitations as the aggregation method used is based on the weighted average rule, which may lead to suboptimality. In [29], the secure and energy aware data gathering technique (SEEDGT) was proposed as a secure and energy-efficient data gathering technique that combines trust, public key algorithms, and compressive sensing (CS) methods to achieve security while maintaining a fair energy load balance in IoT–WSN. The SEEDGT technique is divided into three phases: cluster formation, network operation, and reconfiguration. During the cluster formation phase, energy-efficient and trust-based methods are used to form clusters and select cluster heads. The network operation phase focuses on securing network data during the data-collection process using a public key algorithm. Additionally, the CS strategy is employed to reduce the size of the original data, thus reducing energy consumption. Finally, the reconfiguration phase considers any changes that may occur during network operations. In [30], for data aggregation, the cluster head is selected using the emperor penguin optimization (EPO) algorithm, and STELR is used to ensure routing security. Secure communication is achieved by determining a node’s trustworthiness. In [31], a fog-based hierarchical trust mechanism is proposed, consisting of two components: trust in the underlying structure and trust between cloud service providers (CSPs) and sensor service providers (SSPs). To enhance the trust in the underlying structure, a behavior-monitoring component was established and implemented in wireless sensor networks (WSNs), while the more fine-grained and complex data analysis component was moved to the fog layer. The fog layer focuses on the real-time comparisons of service parameters, gathering exception information in WSNs, performing targeted quantitative evaluations of entities, and performing other aspects to establish trust between CSPs and SSPs. In [32], a trust evaluation model based on trust transitivity along a chain is proposed, which relies on the relatively strong computing and storage capabilities of mobile edge nodes. The authors first designed calculation methods for various trust chains to determine their trust levels. Then, they proposed an improved version of Dijkstra’s algorithm for collecting trust information from sensor nodes by the mobile edge nodes. In [33], a method is proposed for encoding returned verification messages in order to reduce the cost of obtaining verification messages for IoT devices. Additionally, the authors applied an adaptive active trust detection technique that considers the varying energy consumption of IoT devices in order to obtain reliable device trust while preserving network lifetime. The cloud distributes tasks to all available mobile edge units (MEUs). The publication includes the topology of the sensor network and the location of each sensor node. Upon receiving the publication, interested MEUs report their evaluation quality, coverage area, and expected price to the cloud. These three factors form a bid. The cloud then determines the winning set of MEUs and assigns tasks based on the bids. The selected MEUs use their mobile capabilities to collect sensor data and determine if object nodes are malicious. The cloud then aggregates the results reported by the MEUs to obtain the final trust evaluation of each sensor.

Nondominated sorting genetic Algorithm 2 (NSGA-II) [34] and nondominated sorting genetic Algorithm 3 (NSGA-III) [35] are both optimization algorithms used for solving multiobjective optimization problems, where multiple conflicting objectives need to be simultaneously optimized. NSGA-II is an improvement over the original NSGA algorithm and has become a benchmark for multiobjective optimization algorithms. It sorts the population of candidate solutions into different levels of nondomination, creating a front of solutions with a tradeoff between objectives. The algorithm uses the crowding distance measure to maintain diversity among the solutions and prevent premature convergence. NSGA-II has been widely used in various engineering and scientific applications.

NSGA-III, on the other hand, is a further improvement over NSGA-II that addresses some of its limitations. It uses an external archive to store the best nondominated solutions found so far, which is used to guide the search process toward the optimal solutions. NSGA-III also uses a K-means clustering method to divide the population into smaller groups, which allows for a more targeted search and faster convergence. Additionally, NSGA-III has a more flexible problem-specific ideal point, which provides a better reference point for the optimization process. These improvements make NSGA-III a more efficient and effective algorithm for solving multiobjective optimization problems.

A summary of the various methods in terms of different comparison criteria is presented in Appendix A. 

Overall, none of the current algorithms for trust-aware data collection consider the collection of trust using multiple mobile data collectors or the dynamic adjustment of their trajectories for achieving the optimal trust values and network performance aspects such as cost, energy consumption, delay, and quality of service. Due to the partial conflict between the objectives, dealing with this object requires a multiobjective optimization approach.

### 2.2. MCDM for Data Gathering

The second part of the literature survey presents the usage of MCDM models in various WSN–IoT applications. In [36], entropy-based technique for order preference by similarity to an ideal solution is a multiobjective decision-making method (TOPSIS). the field observation instruments networks (FOINs) clustering routing algorithm (ETC) was proposed in this method. Through several variables, the ETC algorithm selects the perfect cluster head (optimal CH). It primarily addresses the issue that certain multiobjective optimization algorithms in use today cannot assign weights in a dynamic and objective manner. Each decision-making criterion for the issue reflects a characteristic for o multi-objective decision-making problem of CHs selection in the clustering routing method. As a result, the decision-making criteria are treated as characteristics and any node in the FOIN having values for the various criteria is chosen as an alternative. 

In [37], the use trusted, energy- and spatial-aware dynamic distance source routing is suggested. The method determines the assessment metrics that are stated in the QoS. In addition, this model introduces a novel hierarchical trust method that uses several features of numerous wireless sensor nodes in accordance with data transfer speed, data size, energy consumption, and suggestions. 

In [38], an MCDM framework was used to resolve challenging routing issues under the assumption of metric threshold limitations. Such needs are captured by the mathematical framework, which also determines the routing.

In [39], a hybrid network architecture for disaster area wireless networks (DAWNs) is proposed. To find the ideal next-hop node in DAWNs, researchers applied a multi-criteria decision-making method for emergency communication protocol (MCDM-ECP) that makes use of the analytic hierarchy process (AHP) method and the technique for order preference by similarity to an ideal solution (TOPSIS) method.

Authors [40] put forward a framework for sensor selection and correlation that allows a reported irregularity in the monitored environment to be activated and verified. It frames and solves the sensor selection problem as a multiobjective optimization (MOO) problem, taking into account the edge gateway device’s free CPU and RAM, network quality, remaining battery life, sensor correlation value, and power requirement of the sensors. In order to validate the anomaly, the sensor nodes are chosen using a multicriteria decision-making process (MCDM)

Authors [41] created a route design technique that is optimized for energy efficiency, extending network lifespan and improving connectivity. The method consists of four steps. The first divides the sensing field into equal sections based on the number of mobile sinks that have been deployed to solve the energy-hole issue. A stable election algorithm (SEA), a heuristic clustering technique, was developed to reduce message transmission between sensor nodes and avoid frequent cluster head rotation. In order to determine the ideal site for the sinks to halt and gather data from cluster heads, a sojourn location determination technique was suggested based on the minimal weighted vertex cover problem (MWVCP). Finally, using multiobjective evolutionary algorithms, three optimization strategies are used to assess the trajectory of the modified mobile sinks (MOEAs). They provided that the multiobjective evolutionary algorithms (MOEAs) may be deemed as an optimum solution for multiple-criteria decision-making (MCDM) problems because MOEAs’ evaluation encompasses many metrics. However, no explicit MCDM algorithm was provided for selecting a solution. 

Overall, it was found in Table 1 that none of the existing approaches in WSNs applies MCDM for enabling decision makers to select one solution out of the set of nondominated solutions in the data gathering based on the factors of trust, effort, cost, and time. Hence, we aimed filling this gap by using the simple additive weighting (SAW) from MCDM for this purpose. 

## 3. Methodology 

Almost all of the symbols in the paper are represented in Table 2.

### 3.1. Definitions

This section presents several definitions used for building our methodology. 

#### 3.1.1. Definition 1: Class

A class is a portion that represents the set of all same size and types solutions in the problem space. More formally, for all x∈ class C, then $length\left(x\right)=N_{C}$, where $N_{C}$ denotes the length of solutions inside class C, and $type(x_{i})$ is the same as $type(y_{i})$ for two decision variables $x_{i}$ and $y_{i}$, belonging to x and y, respectively, where x and y are from the same class. 

#### 3.1.2. Definition 2: FAM

The matrix of fitness adjacencies FAM represents a matrix with many columns equal to the number of objective functions and many rows equal to the number of solutions. The fitness value of the solution with each objective function is represented by an entry in the matrix. When displaying the solution’s fitness to the different objective, FAM is useful. In the equation, FAM is a represented by.
(1)$$FAM=\left(\begin{array}{c} \begin{array}{ccc} {fam}_{11} & . & \begin{array}{ccc} . & . & {fam}_{1n} \end{array} \end{array}\\ \begin{array}{ccc} {fam}_{21} & . & \begin{array}{ccc} . & . & {fam}_{1n} \end{array} \end{array}\\ \begin{array}{c} \begin{array}{ccc} . & . & \begin{array}{ccc} . & . & . \end{array} \end{array}\\ \begin{array}{ccc} . & . & \begin{array}{ccc} . & . & . \end{array} \end{array}\\ \begin{array}{ccc} {fam}_{mn} & . & \begin{array}{ccc} . & . & {fam}_{mn} \end{array} \end{array} \end{array} \end{array}\right)$$
where ${fam}_{ij}$ denotes fitness solution j with regard to objective i.

#### 3.1.3. Definition 3: RAM

The reduced adjacency matrix (RAM) is a matrix with a row count equal to the number of classes in the solution space and a column count equal to the number of objectives. Concerning the appropriate objective function, each entry in the matrix denotes an operator that is applied to the class’s solutions. The average, minimum, maximum, and median are a few examples of possible operators.
(2)$$RAM=\left(\begin{array}{c} \begin{array}{ccc} {ram}_{11} & . & \begin{array}{ccc} . & . & {ram}_{1n^{\prime }} \end{array} \end{array}\\ \begin{array}{ccc} {ram}_{21} & . & \begin{array}{ccc} . & . & {ram}_{2n^{\prime }} \end{array} \end{array}\\ \begin{array}{c} \begin{array}{ccc} . & . & \begin{array}{ccc} . & . & . \end{array} \end{array}\\ \begin{array}{ccc} . & . & \begin{array}{ccc} . & . & . \end{array} \end{array}\\ \begin{array}{ccc} {ram}_{mn} & . & \begin{array}{ccc} . & . & {ram}_{mn^{\prime }} \end{array} \end{array} \end{array} \end{array}\right)$$
where ${ram}_{ij}$ denotes the fitness value of class j concerning objective i.

There are several approaches to depict the fitness value of a given class. We focused on these three: 1. the class’s lowest fitness value for solutions; 2. the class’s average fitness score; 3. the subject class’s set coverage in comparison with that of other classes.

#### 3.1.4. Definition 4: WCT

The weak classes threshold (WCT) is a measure of how many solutions are used in a given class to make it a weak class. When a class has fewer solutions than the WCT, the class becomes weak.

#### 3.1.5. Definition 5: SCT

The strong class threshold (SCT) is a measure of how many solutions are utilized to make a certain class a strong class. When the class has more solutions than the SCT, the class becomes a strong class.

### 3.2. Overview of SC-MOPSO

The pseudocode of SC-MOPSO is presented in Algorithm 1. As shown, the inputs of the algorithms are NumOfParticles, which indicates the number of particles; NumOfIterations, which indicates the number of iterations; Boundary, which identifies the outermost search; DimensionsRange, which indicates the range of dimension; ObjectiveFunctions, which indicates to the objective functions that need to be optimized. In addition, the parameters of the traditional particle swarm optimization, namely, inertia, $C_{1}$, and $C_{2}$ are given. Furthermore, the algorithm accepts two thresholds, namely, adaptive timeout and classMinThreshold. The former indicates the threshold that allows moving a particle from one class to another, and the latter indicates the minimum number of particles within a class. The outputs of the algorithm are the Pareto front and Pareto set. 

The particle distribution among the classes is the algorithm’s foundation. The first step in the distribution of the particles involves employing two random number generators: one to assign each particle in class and the other to generate the values of the decision variables of the particle. In this context, the term “class” refers to a subset of the solution space that only includes solutions from the same dimension and type of decision variable; all other dimension solutions are found in other classes.

The pseudocode for Algorithm 1 presents the first swarm initialization phase (lines 3 to 8). The solutions are projected into their classes using the command on line 8. The algorithm then executes for the specified number of iterations (NumOfIterations). The algorithm runs over each class one by one with each iteration. The algorithm runs through each class’s particles one at a time and uses the function selectExemplar (Classes) to choose an exemplar for each particle. Any particle’s exemplar is chosen from the same class as the particle. The particle is then moved to a new position by the algorithm depending on its input parameters: (Inertia, C1, and C2) and exemplar. The algorithm evaluates any potential improvement in the fitness functions after transferring the particle. Afterward, the algorithm adds one to the nonimprovement counter if there is no improvement; otherwise, the improvement counter is reset. The improvement counter’s purpose is to make it possible for the logic to move the particle from one class to another if no improvement happens after a certain number of repetitions, which is represented by the adaptiveTimeOut threshold. A minimal number of particles must exist within a specific class in order for logic to move particles from one class to another. This is represented by another threshold, which is classMinThreshold.
**Algorithm 1** Pseudocode of SC-MOSPO for the V-length problem [42]**Input**1-NumOfParticles2-NumOfIterations 3-Boundary 4-DimensionsRange 5-ObjectiveFunctions 6-Inertia,C1,C27-adaptiveTimeOut 8-classMinThreshold**Output**1-ParetoFront2-ParetoSet **Start Algorithm**1: initSwarm(NumOfParticles)2: numOfClass = DimensionsRange.Max − DimensionsRange.Min + 1;3: for i = 1 until NumOfParticles4:    Dimension = genRan(DimensionsRange)5:    particle = genRan(Dimension,Boundary)6:    add particle to swarm 7: end8: Classes = Distribute(swarm)9: for iteration = 1 until NumOfIterations10:  for classIndex = 1 until length(Classes)11:      for each particle of Classes(classIndex)12:      exemplar = selectExemplar(Classes(classIndex)//-1-13:      newParticle = moveParticle(particle,exemplar,Inertia,C1,C2)14:       if(NoImprove(newParticle,particle))15:          particle.counter = particle.counter + 1;16:       else17:          particle.counter = 0;18:          particle = newParticle;19:       end20:      end21:      end22:      for classIndex = 1 until length(Classes)23:      for each particle of Classes(classIndex)24:       adaptiveTimeOut = timeoutUpdate(particle.nonImpWindow, maxTimeOut)25:       if(particle.counter > adaptiveTimeOut and length(particle. Class) > classMinThreshold)26:          Class = selectNewClass(particle) 27:          particle = moveToNewClass(particle, Class) 28:       end 29:     end30:      end31:  end32: end**End Algorithm**


The existing variant of SC-MOPSO assumes that the solution is homogenous. In other words, it considers that the definition of the decision variable is the same for all components of the solution vector. However, this does not apply to the problem of trust-aware data gathering. In such a problem, the decision variable is different from one component to another, i.e., some components indicate the number of mobile sinks, others indicate the number of rendezvous points, others indicate the number of hops, and others indicate their locations. To resolve this problem, we developed a novel variant of SC-MOPSO. We named it SC-MOPSO for data gathering or SC-MOPSO-DG. It provides application-dependent operators named interclass operators. The operators include solution generation, addition and deletion of rendezvous points, moving to an upper class, and moving to a lower class with an awareness of the representation of data gathering solution. This is presented in the next subsection. 

### 3.3. SC-MOPSO-DG 

This section presents our novel algorithm SC-MOPSO-DG. First, we present the mathematical model of the objective space in Section 3.3.1. It is composed of the initialization, which is presented in Section 3.3.2. Next, the adding and deletion of RendezVous point RV is presented in Section 3.3.3. Then, the algorithm for moving to an upper class is presented in Section 3.3.4, and the algorithm for moving to a lower class is given in Section 3.3.5. Next, the main algorithm is presented in Section 3.3.6. Lastly, simple additive weighting (SAW) is presented in Section 3.3.7. 

#### 3.3.1. Mathematical Model of Objective Space 

For conducting optimization, we need to mathematically model each of the objectives, namely, trust, energy, cost, and time using the decision variables. The decision variables are defined using Equation (3):(3)$$x=(Nms,N_{Rv}^{i},{NH}^{i,j})$$
where Nms denotes the number of mobile sinks; $N_{Rv}^{i}$ denotes the number of rendezvous points for mobile sink i; ${NH}^{i,j}$ denotes the number of hops for mobile sink i at rendezvous point j.

(1)Trust

In a wireless sensor network, the trust of a sensor can be defined as the degree of confidence that can be placed in the accuracy, reliability, and security of the data generated by that sensor. A high degree of trust in a sensor implies that the data it generates are accurate, reliable, and secure, while a low degree of trust implies the opposite.

The trust of a sensor can be quantified as an integer value between 1 and 10 based on a number of factors: 1, sensor calibration: a sensor that is properly calibrated and periodically maintained is more trustworthy than one that is not; 2, sensor accuracy: a sensor that consistently generates accurate data is more trustworthy than one that generates inaccurate data; 3, sensor reliability: a sensor that consistently operates without interruption is more trustworthy than one that frequently fails; 4, security: a sensor that is secure and protected against tampering is more trustworthy than one that is not; 5, data integrity: a sensor that generates data that are consistent with other sources of information is more trustworthy than one that generates inconsistent data; 6, age and usage: a new sensor that has not been extensively used is generally more trustworthy than an older sensor that has been in use for a long time.

Based on these and other factors, the trust of a sensor can be quantified as an integer value between 1 and 10, with a value of 1 indicating very low trust and a value of 10 indicating very high trust. This quantification provides a simple, intuitive way to express the degree of trust in a sensor and can be used to inform decisions about how to use and rely on the data generated by that sensor within the wireless sensor network.

Trust is calculated based on the number of mobile sinks, the set of a rendezvous points for each one, and the number of assumed hops for each rendezvous point. We also assume that increasing the number of hops implies obtaining trust, which is the minimum among all hops, as provided in Equation (4).
(4)$$\sum_{i=1}^{Nms}\left(\sum_{j=1}^{Nrv}\left(\frac{T_{j1}^{i}+min\left(T_{j_{1}}^{i},T_{j_{2}}^{i}\right)+min\left(T_{j_{1}}^{i},T_{j_{2}}^{i},T_{j_{3}}^{i}\right)+\cdots +\min\left(T_{j_{1}}^{i},T_{j_{2}}^{i},T_{jNhop}^{i}\right)}{Nrv}\right)\right)$$
where $T_{jk}^{i}$ denotes the trust of sensor that sending data to mobile sink i at RV j from hop index k, $T_{jk}^{i}\in \{1,10\}$.

(2)Energy Consumption

Second, the energy consumption is calculated based on the energy at each sink and rendezvous point as described in Equation (5).
(5)$$Energy=\sum_{i=1}^{Nms}\;\sum_{j=1}^{Nrv^{i}}\;\left(\sum_{k=1}^{N^{i,j}}\;E_{i,j,k}+\sum_{k=1}^{N^{j,1}}\;E_{i,j,k}+\cdots \ldots +\sum_{k=1}^{N^{j,NHops}}\;E_{i,j,k}\right)$$
where $Nrv^{i}$ denotes the number of rendezvous points for mobile sink i; $N^{i,j}$ denotes the number of sensors at the rendezvous point j for mobile sink i; $E_{i,j,k}$ denotes the energy consumption at sensor k, rendezvous point j, and mobile sink i.

(3)Cost

Third, the cost is calculated as the number of mobile sinks. Fourth, the time is calculated as the longest travelling time among all mobile sinks, as described in Equation (6).
(6)$$Cost=A_{1}N_{ms}+A_{2}$$

(4)Time

The time indicates the longest mobile sink in terms of collecting data from its associated sensors, as described in Equation (7).
(7)$$Time=\max\left(d_{1},d_{2},{\ldots .,d}_{N_{ms}}\right)$$
where $d_{i}$ denotes the distance travelled by mobile sink i, and it is calculated based on the summation of the distance between every two consecutive distances, as provided in Equation (8).
(8)$$d_{i}=\sqrt{\sum_{J=2}^{N_{Rv},i}{\left({RP}_{j}-{RP}_{j-1}\right)}^{2}}$$
where ${RP}_{j}$ denotes the rendezvous point. 

(5)Noncoverage

Noncoverage indicates to the number of sensors that have not been covered by the mobile sink. It is calculated based on subtracting the number of connected sensors Nconn from the total number of sensors Ns, as described in Equation (9).
(9)NonCoverage=Ns−Nconn

#### 3.3.2. Initialization

The solution representation is heterogeneous because it combines different types of decision variables. Therefore, the process of constructing solutions should be carefully approached to ensure that the solutions are distributed over the range of each variable. To initialize the process, we use a random process while also considering the boundary information of each variable to ensure that each solution does not violate any constraints. The constraints are defined based on the boundary information given in several variables: 1, lower and upper boundary of positions ${BP}_{L}$ and ${BP}_{H}$ respectively; 2, lower and upper boundary of other decision variables ${BD}_{L}$ and ${BD}_{H}$, respectively; 3, index of number mobile sink boundaries within ${BD}_{L}$ and ${BD}_{H}$, and it is denoted by IMS; 4, index of number of rendezvous points boundaries within ${BD}_{L}\mathrm{and}{BD}_{H},$ and it is denoted by IR; 5, index of number of hops boundaries within ${BD}_{L}\mathrm{and}{BD}_{H},$ denoted by INH.

The pseudocode is presented in Algorithm 2. As shown, the solution is created as a structure. Next, a random process is used for initializing the number of mobile sinks. Afterward, a for loop is conducted on the mobile sink, and other information is generated including the number of rendezvous points, the location of each rendezvous point, and the number of hops using an internal loop.
**Algorithm 2** Initialization of SC-MOPSO-DG***Inputs***1-BPH 2-BPL 3-BDH 4-BDL **Output:**1-Sol 2-Class 3-Dim **Start Algorithm**1: Sol = struct()2: Sol.ms = struct()3: Nms= randi(BDL (IMS)) BDL(IMS))]) 4: class= Nms 5: dim = 06: for i = 1:Nms do 7:  solution.ms(i).NRP = struct()8:  Nrv = randi([BDL(IRV) BDL (IRV)])9:  for j = 1:Nrv do 10:   solution.ms(i).NRP(j).pos = [rand*(BPH (XI) − BPL (XI) − rand*(BPH (YI) − BPL (YI))] 11:   solution.ms(i).NRP(j).nHops = randi([BDL(IRV) BDH (IRV)]) 12: End for 13: dim = dim + Nrv*length of one RV information14: class = strcat(num2str(class),num2str(Nrv)) 15: end for**End Algorithm**

#### 3.3.3. Addition and Deletion of Rendezvous Point 

An issue with moving a solution from one class to another is the need to add more information to the solution when the destination class is higher or to delete older information from the solution when the destination class is lower. To handle this, an algorithm for addition and deletion was developed. The algorithm receives the destination class as the class input, an exemplar that represents a leader selected from the destination class; and OldSolution, which represents the subject solution that is being moved from the source class to the destination class. 

As shown, the Algorithm 3 starts operating by initiating newSolution, which represents the solution after being moved to the destination class. First, newSolution takes all the information of the oldSolution. Next, assuming that all solutions have the same number of mobile sinks, the differences between the number of rendezvous points in the oldSolution and in the corresponding newSolution is calculated. There are two possibilities: (1) the difference is negative, which means that rendezvous points are to be added; (2) the difference is positive, which means that rendezvous points are to be deleted. In the first case, we randomly select one rendezvous point from the exemplar and add it to the solution until equality. In the second case, we select one rendezvous point from the subject solution and delete it until the difference is 0, which means that the solution is already transformed to the new class.
**Algorithm 3** Pseudocode of adding and removing rendezvous points**Input**1-class: class for moving to it.2-exemplar: individual solution of new class.3-oldSolution: the particle needed to moved.**Output**1-newSolution**Start Algorithm**1: newSolution = oldSolution2: for each ms of newSolution do3:  difference = number of RV(ms) of exemplar − number of RV(ms)4:  if difference > 0 then5:   for i = 1 until difference do6:     select RVs randomly from exemplar7:     add them to ms newSolution8:   end for9:   else if difference < 0 then10:   for i = 1 until difference do11:     delete RV randomly12:   end for13:  end if14: end for**End Algorithm**

#### 3.3.4. Moving to a Higher Class

Assume that we have an oldParticle, exemplar, class, and difference. Algorithm 4 plays the role of adding the same number provided in the difference in mobile sinks to the oldParticle. In other words, we select a random mobile sink from the exemplar and add it to the oldParticle, then we call the adding and deleting of the rendezvous point until the number of newly added mobile sink values is equal to the difference. Hence, in this case, we consider that the solution has been moved from a lower class to an upper class.
**Algorithm 4** Move to higher class**Input**1-class: class for moving to it.2-exemplar: individual solution of new class.3-oldParticle: the particle needed to moved.4-difference: difference between the number of mobile sinks of old and new class particle **Output**1-newParticle.**Start Algorithm**1: newParticle = oldParticle2: for i = 1 until difference do3:  select random ms(exemplar) and add it to newParticle4: end for5: newParticle = AddDeleteRVs(newParticle, exemplar, class)**End Algorithm**

#### 3.3.5. Moving to a Lower Class 

Assume that we have an oldParticle, exemplar, class, and difference. Algorithm 5 plays the role of detecting the same number provided in the difference in the mobile sinks from the oldParticle. In other words, we select a random mobile sink from the oldParticle, then we call the deletion of the rendezvous point until the number of deleted mobile sink value is equal to the difference. Hence, in this case, we consider that the solution has been moved from an upper class to a lower class.
**Algorithm 5** Move to a lower class**Input**1-class: class for moving to it.2-exemplar: individual solution of new class.3-oldParticle: the particle needed to moved.4-difference: difference between the number of mobile sinks of old and new class particle.**Output**1-newParticle.**Start Algorithm**1: newParticle = oldparticle2: for i = 1 until difference) do3:  delete random ms from newParticle4: end for5: newParticle = AddDeleteRVs(exampler,newParticle,class)**End Algorithm**

#### 3.3.6. Main Algorithm

To accomplish the movement function of social class MOPSO DG, we use the pseudocode provided in the main algorithm. As shown, the algorithm uses the standard movement equation of particle swarm optimization that is provided in Algorithm 6 on line 13. 

The pseudocode of the algorithm is presented in Algorithm 6. It accepts the setupParameters as inputs, and it provides the Pareto front as the output. The algorithm starts with the initialization of the population, evaluating the population based on objective functions, determining nondominant solutions, creating the repository, and initializing the AMC and RAM matrices. Next, the algorithm enters a set of iterations until reaching the quantity of iterations. Each time the algorithm iterates, it updates the parameters, then considers the classes one by one, selects a random exemplar within the class, moves the solutions toward their exemplar according to the mobility equations of the algorithm, and checks for crossing the boundary and fixing the solutions, in this case, to be maintained within the boundaries. Afterward, it provides the mutation and updates the solution in case of domination in the new position. In addition, it updates the improvement counter. 

The second part of Algorithm 6 considers the classes again and updates the enhancement time out. Next, it considers the solution of each class and checks whether the solution has reached its timeout within the class without improvement. In this case, the solutions are moved from their present class to a different one. This leads to calling the process of moving, which has two functions: (1) moving to the upper class if the destination class has more mobile sinks and (2) moving to the lower class if the new class has a smaller number of mobile sinks. Next, the matrices of FAM and RAM are updated. 

Our developed variable-length multiobjective SC-MOPSO-DG optimization algorithm is presented in Algorithm 4. It accepts the solution dimension d, the population size n, the max iteration, and the objective functions. It returns the Pareto front. It operates by starting with an initialization of the population according to its size. Next, it evaluates the member of the population, and it updates the values of FAM and RAM. Afterward, it performs nondominated sorting based on the pseudocode presented in Algorithm 4. Next, it enters the for loop that has a maximum equal to the maximum number of iterations Each time the algorithm iterates, it selects an exemplar for each class, applies the movement equations of SC-MOPSO-DG optimization, and corrects the solution when exceeding the boundary of the solution space. Next, the algorithm updates the improvement counter of each solution. For the nonimproving solutions, the algorithm moves them from their class to another class. Lastly, the algorithm deletes weak classes. Lastly, the algorithm calls the archive controller algorithm and updates FAM and RAM.
**Algorithm 6** Pseudocode of SC-MOPSO**Input**1-d solution dimension2-n population size 3-max_iteration4-f=[f1X1f2X1 …fmX1] multi objective functions **Output**1-Pareto front **Start Algorithm**1: Initialize the SC-MOPSO-DG population based on the number n2: Evaluate the fitness of population using objective functions 3: It updates FAM and RAM matrices 4: Perform nondominated sorting on the population 5: Determine nondominated solutions and add them to archive using archive controller 6: For i = 1 until max_iteration7:  select exemplar from each class based on FAM based probability density function8:  Update particle positions using algorithm Equations (1)–(4) and selected exemplars 9:  Check if any search agent goes beyond the search space and amend it10:  Calculate fitness values of solutions,11:  update improvement counter for each solution12:  move weak solutions from their class to other class in case they are not improving13:  deletes weak class 14:  call archive controller and update FAM and RAM15: Return archive **End Algorithm**

#### 3.3.7. Simple Additive Weighting (SAW) 

Simple additive weighting (SAW) is one approach as in Figure 1 to the multiattribute decision-making problem. The fundamental idea behind the SAW approach is to calculate the average of the weighted performance ratings for each option across all criteria. It calls for a procedure to convert the choice matrix (X) to a scale that can be compared with all the ratings of available options. In our optimization, we have four criteria, namely, cost, energy, time, and trust. 

We extract the solutions from the Pareto front (PF), and we provide them to table the PF. Next, we require the decision maker to provide the weights of the criteria according to their preferences. We normalize the matrix, and we perform the ranking based on the weighted sum. The solution that accomplishes a higher rank is selected. The weights of the criteria are given in Table 3, the Pareto front is given in Table 4, and the overall evaluation is given in the Pareto front in Table 4. The process of SAW is depicted in Figure 2, The ranking of solutions provided in the Table 5.

We present the pseudocode of selecting the solution from the Pareto front in Algorithm 7.
**Algorithm 7** Pseudocode of selecting solution from Pareto front using simple additive weight (SAW)**Input:**paretoFront W objectives’ weights **Output:**
objectivesOptFuncs **Start: **
objectivesFuncs = paretoFront.solutionsObjectiveValues minValue = min(objectivesFuncs) maxValue = max(objectivesFuncs) normObjectives = (objectivesFuncs − minValue)/(maxValue − minValue) sumWObj = sum(normObjectives.*W) [minValues,index] = min(sumWObj) objectivesOptFuncs = paretoFront.solutionsObjectiveValues(index) **End**


### 3.4. Example

Assume that we have two solutions, ${sol}_{1}$ and ${sol}_{2}$. The first solution ${sol}_{1}$ contains two mobile sinks, namely, ms(1,1) and ms(1,2). The second solution ${sol}_{2}$ contains three mobile sinks, namely, ms(1), ms(2), and ms(3). Each mobile sink has its own number of rendezvous points. For the mobile sink ms(1,1), we have three RVs, namely, [150 400], [200 144], and [160 350]. In addition, each RV has its own number of hops. The number of hops of RV(i,j,k) is 2 for i=j=k=1, as represented in Table 6.

### 3.5. Performance Metrics

For evaluating our developed algorithm, we generated two types of metrics: multi-objective optimization (MOO) and application-oriented metrics. For the first type, we calculated the hyper-volume, set coverage, delta, and the number of nondominated solutions. For the second type, we calculated time-series-based metrics for the packet delivery ratio (PDR), end to end delay (E2E-delay), and energy consumption. The equations of the second type are presented in Section 5 while the equations of the first type are presented in this section. 

The effectiveness of search algorithms was assessed using the hyper-volume (*HV*) metric in evolutionary and swarm multiobjective optimization. In relation to the poorest solution (the reference point), it calculates the volume of the dominated area of the objective space; this region is the union of the hypercube whose diagonal is the separation between the reference point and a solution *x* from the Pareto set (*Ps*). More preferable solutions are indicated by higher values of this measure indicator, as shown in Equation (10).
(10)$$HV=volume\left(U\;HyperCube(x)\right)$$

The set coverage, or *C* metric, which compares the Pareto sets *Ps*1 and *Ps*2, is shown in Equation (11).
(11)c(Ps1,Ps2)=|{y∈Ps2|∃x∈Ps1:x<¬y}| |Ps2|

*C* is equal to the proportion of *Ps*2’s nondominated solutions to *Ps*1’s nondominated solutions divided by *Ps*2’s total number of solutions. Therefore, it is crucial to reduce the value of *C*(*X*, *Ps*) for all Pareto sets *X* while assessing a set of *Ps*.

## 4. Experiment and Results

This section presents the experimental methods and results. It consists of two subsections, namely, MOO evaluation, which is presented in Section 4.1, and networking measures, which are presented in Section 4.2. The generation of the experimental results was performed using MATLAB 2020b using a computer i7 core 2.30 GHZ 16GB RAM. The parameters that were used for conducting the experimental evaluation are presented in Table 7. As shown in the table, we used 100 iterations for all algorithms. Furthermore, the boundary dimension of all algorithms ranged between [1,2,2] and [3,4,6], which indicates the range of [number of mobileSink, number of RV, number of hops]. 

### 4.1. MOO Evaluation

This section presents our experiments and results. We present the set coverage to show the domination performance of the benchmarks and of our created algorithm. A comparison is provided with two benchmarks, namely, NSGA-II and NSGA-III. Afterward, we present the hyper-volume, which shows diverse behavior. Lastly, we present the number of nondominated solutions. 

#### 4.1.1. Set Coverage 

As shown in Figure 3, the domination of our developed algorithm with those of each of NSGA-II and NSGA-III is presented. It was observed that the set coverage of SC-MOPSO was 0.06 over NSGA-II compared with only a domination of 0.04 of NSGA-II over SC-MOPSO. 

The set coverage of SC-MOPSO with NSGA-II is presented in Figure 3. There is a blue rectangle between Q1 and Q3. The maximum and minimum are represented in black. The median is represented in red line. The results show more domination of SC-MOPSO over NSGA-II. The domination percentage in terms of median value is almost 0.06. On the other side, the domination of NSGA-II over SC-MOPSO is approximately 0.025 in terms of the median value. Similarly, we present the set coverage of SC-MOPSO with NSGA-III in Figure 4. The results show equivalent domination in terms of the median value with value close to 0.01. This provides that the domination is almost the same between the two algorithms. However, there is more interquartile range for the domination of NSGA-III over SC-MOPSO due to the higher distance between minimum and maximum. 

#### 4.1.2. Hyper-Volume

The hyper-volume results are presented in Figure 5. We observed that SC-MOPSO slightly outperformed NSGA-III in terms of hyper-volume, while NSGA-II produced a result of approximately 2.4 $\times 10^{4}$. Reading these figures together with that for set coverage showed that SC-MOPSO outperformed NSGA-II in terms of set coverage, while it outperformed NSGA-III in terms of hyper-volume. 

#### 4.1.3. Number of Nondominated Solution

The number of nondominated solutions is presented in Figure 6. All algorithms generated the same level of number of nondominated solutions (NDSs). The size of the population for each algorithm was 200. This led to a number of nondominated solutions of 200. This was interpreted with the high number of objectives, which led to the high number of nondominated solutions. 

#### 4.1.4. Convergence Graphs

For additional evaluation, we present a boxplot convergence curve of each of the objectives with respect to the generation from the starting point of the optimization until the end. We can observe from the figures that SC-MOPSO had fewer outliers than the other two algorithms, namely, NAGA-II and NSGA-III. This was generalizable to the trust results in Figure 7.

Contrary to expectations, the energy consumption profile decreased as the number of iterations increased from 10 to 100 for all three algorithms. The energy consumption for each algorithm started at 0.05 and decreased as the number of iterations increased. A similar pattern can be observed for all three algorithms in Figure 8. However, both NSGA-II and NSGA-III had more outliers than SC-MOPSO-DG.

Contrary to expectations, the cost profile decreased as the number of iterations increased from 10 to 100 for all three algorithms. The cost for each algorithm started at 0.05 and decreased as the number of iterations increased. A similar pattern can be observed for all three algorithms in Figure 9. However, both NSGA-II and NSGA-III had more outliers than SC-MOPSO-DG.

The outcomes of our study are depicted in Figure 10, where we compare the performance of our newly developed SC-MOPSO-DG algorithm with that of two existing benchmarks. The results demonstrate that the median value of SC-MOPSO-DG significantly improved and reached a level below 1500, which is comparable to that of the other two benchmarks. Moreover, we observed that NSGA-II, one of the benchmark algorithms, had a higher number of outliers and incurred a greater cost than the other two algorithms. Overall, these findings indicate that SC-MOPSO-DG is a promising algorithm that can achieve comparable or better results than the existing benchmarks while being less susceptible to outliers and reducing the overall cost of the optimization process.

The final metric that we analyzed was noncoverage, which is shown in Figure 11. The results indicate that the noncoverage of SC-MOPSO-DG significantly decreased to 40%, which is much lower than that of NSGA-II (48%) and NSGA-III (62%). This finding suggests that SC-MOPSO-DG outperforms the other two algorithms in terms of noncoverage, which is a desirable characteristic in multiobjective optimization. Therefore, our results demonstrate the superiority of SC-MOPSO-DG over the other two benchmarks and highlight its potential for addressing complex optimization problems that involve multiple objectives.

### 4.2. Networking Measures

This section presents the networking evaluation. It is decomposed into four parts, namely, E2E delay, energy consumption, PDR, and trust. 

#### 4.2.1. E2E Delay

The time series is presented in Figure 12 of the E2E delay shows that the best performance was achieved by SC-MOPSO, which produced a delay with level 0.3; NSGA-II and NSGA-III generated a higher E2E delay. In addition, we observed that the behavior of SC-MOPSO produced a transient state that slowly increased until reaching a steady state case at 0.3, while NSGA-III produced a different pattern of the transient state, which started high and slowly decreased until reaching the steady state of 0.5 s. 

#### 4.2.2. Energy Consumption

The energy consumption is presented in Figure 13. We found that the solution provided by SC-MOPSO generated a higher level of energy consumption with an increasing pattern than NSGA-II and NSGA-III. This is consistent with the self-conflicting nature of objectives.

#### 4.2.3. PDR

The other metric that was calculated was the packet delivery ratio (PDR) for the solutions generated from our developed method and the two benchmarks. As shown in Figure 14, SC-MOPSO generated the highest PDR, reaching a value of 100% compared with a PDR of around 90% for NSGA-II and of around 40% for NSGA-III.

#### 4.2.4. Trust

The last metric that was calculated was trust, and it is given in Figure 15. The results show that trust reached a value of 10 for NSGA-II, which is higher than the around 8 for NSGA-III and around 6 for SC-MOPSO. However, considering that we selected a solution from the Pareto front, it is normal for one objective to be dominant over other objectives.

## 5. Discussion

The problem of gathering data through the use of multiple mobile sinks, multiple rendezvous points, and various numbers of hops is a multiobjective problem. As a result, two types of metrics were used to compare the results. The first type was MOO metrics, where set coverage, hyper-volume, number of nondominated solutions, and convergence graphs were utilized. Networking metrics, including E2E delay, energy consumption, PDR, and trust, were also calculated.

Regarding the first type, the results demonstrated that SC-MOPSO-DG is superior to NSGA-II in terms of most metrics and comparable to NSGA-III in terms of some metrics, while being superior to NSGA-III in terms of hyper-volume. For example, it was competitive with NSGA-III in terms of domination and was superior in terms of hyper-volume.

Regarding the second type, we only chose one solution from the Pareto front of each algorithm and generated the corresponding time series of the solution. The results were superior to those of the benchmarks in terms of PDR and E2E delay, demonstrating an excellent quality of service. Additionally, it showed a higher energy consumption profile and less trust, which is normal given the domination aspect of the solutions in the Pareto front and the self-conflicting nature between the objectives. The overall superiority of our developed SC-MOPSO-DG is attributable to the developed operators for class interaction, which enables moving solutions among various classes and consequently enabling variable-length searching.

## 6. Conclusions and Future Work

In a WSN, sensor nodes are typically deployed to monitor the environment and collect data. Trust-aware data gathering involves determining which nodes can be trusted to reliably collect and transmit data. Social class assignment is a mechanism used to classify sensor nodes into different groups based on their trustworthiness or reliability. Nodes in higher social classes are considered more reliable and are given higher priority in data-gathering tasks. The proposed algorithm addresses the challenge of finding an optimal social class assignment for sensor nodes. It introduces a variable-length encoding scheme to represent the social class assignment, allowing for flexibility in the number of social classes and their sizes. This enables the algorithm to adapt to different network conditions and requirements. The optimization problem is formulated as a multiobjective problem, simultaneously considering multiple conflicting objectives. Typical objectives in trust-aware data gathering include maximizing data reliability, minimizing energy consumption, and balancing the network load. The algorithm uses a combination of evolutionary particle swarm optimization, solution generation, and addition and deletion of rendezvous points to enable moving to an upper class or a lower class. Once the algorithm converges, it provides a set of multiobjective optimization metrics, namely, set coverage, hyper-volume, and the number of nondominated solutions, representing different trade-offs between the objectives followed by statistical analysis and selecting a suitable solution based on the specific requirements and priorities of the WSN. In an effort to confirm the proposed design’s performance, a simulation was run to generate the time series of E2E delay, packet delivery ratio, trust, and energy consumption for each of the proposed designs. The SC-MOPSO set coverage was 0.06 over NSGA II compared with only a domination of 0.04 of NSGA-II over SC-MOPSO, while it showed competitive performance with NSGA-III. We concluded that VL-MOSCO is a promising approach for optimizing social class assignment in trust-aware data gathering in wireless sensor networks. By considering multiple objectives and using variable-length encoding, it offers flexibility and adaptability to different network conditions, ultimately improving the overall performance and reliability of the network.

## Figures and Tables

**Figure 1 sensors-23-05526-f001:**
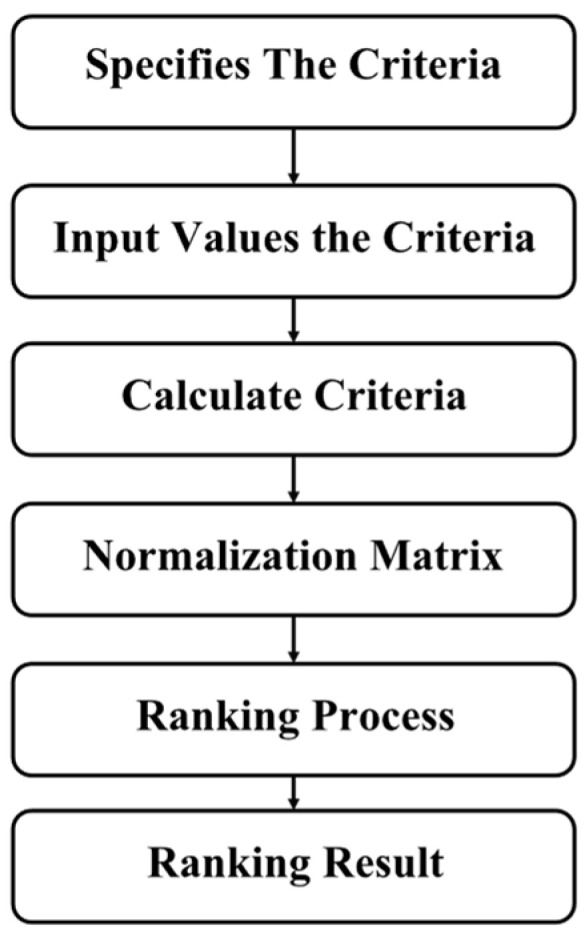
Simple additive weighting chart.

**Figure 2 sensors-23-05526-f002:**
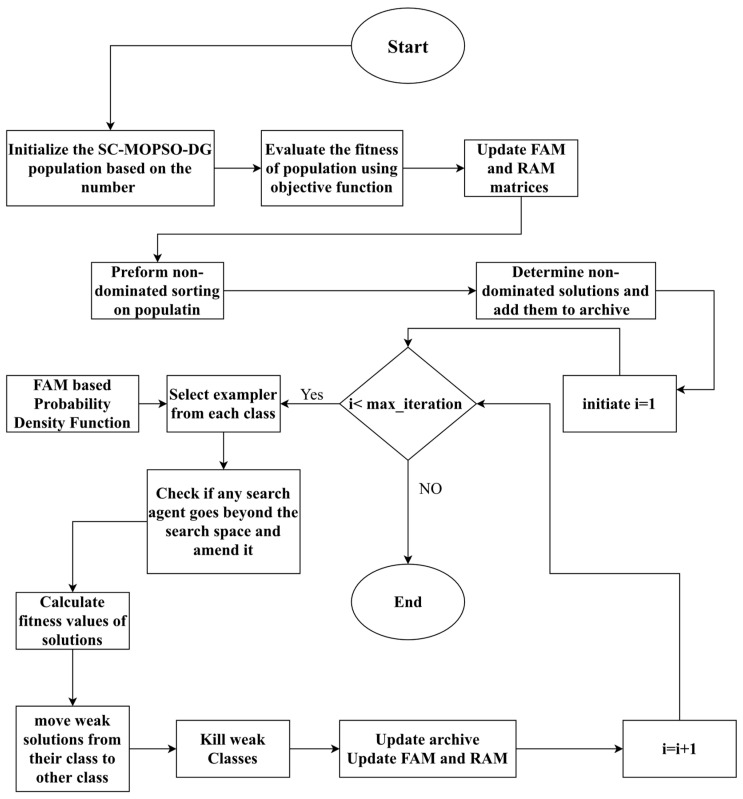
Flowchart of social class multiobjective particle swarm optimization data-gathering algorithm.

**Figure 3 sensors-23-05526-f003:**
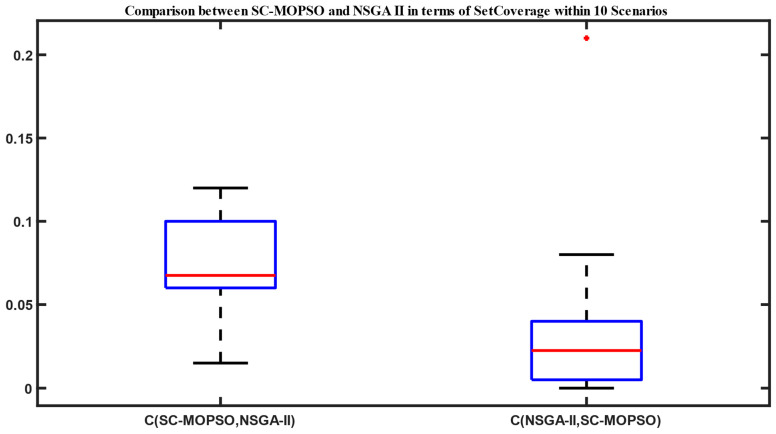
Set coverage comparison between NAGA-II and SC-MOPSO.

**Figure 4 sensors-23-05526-f004:**
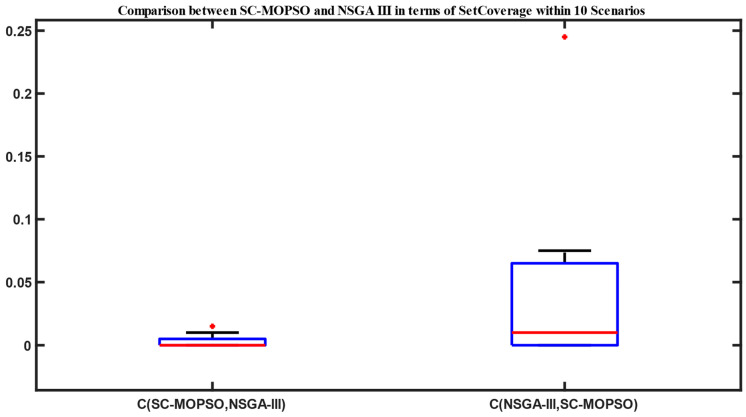
Set coverage comparison between NAGA-III and SC-MOPSO. Red is median, Blue is Q1 and Q3. Black is minimum and Maximum.

**Figure 5 sensors-23-05526-f005:**
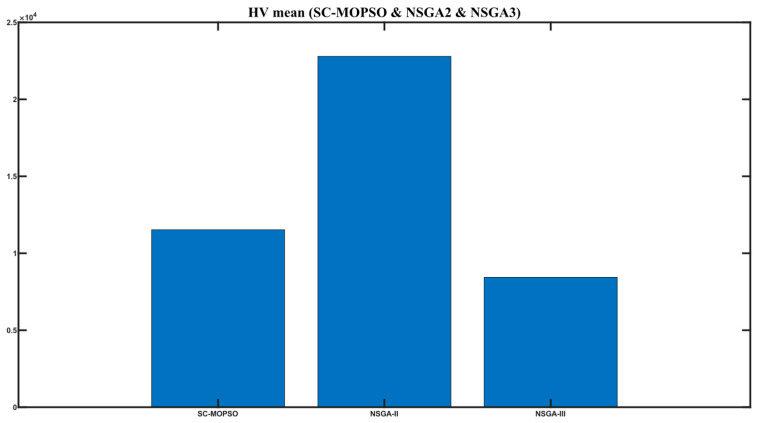
Hyper-volume results of SC-MOPSO and the benchmarks.

**Figure 6 sensors-23-05526-f006:**
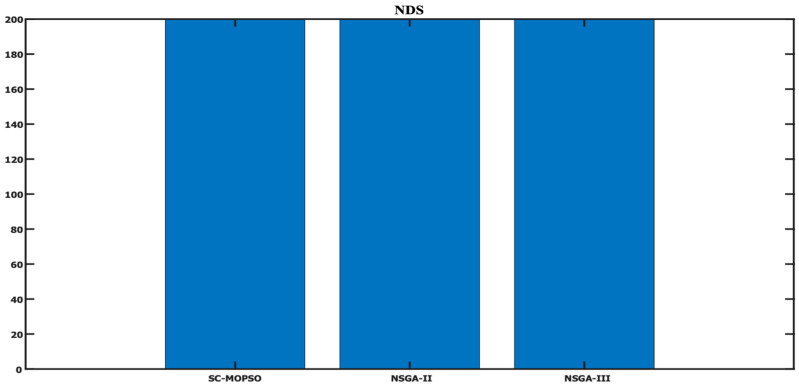
Comparing SC-MOPSO with benchmarks in terms of the number of nondominated solutions.

**Figure 7 sensors-23-05526-f007:**
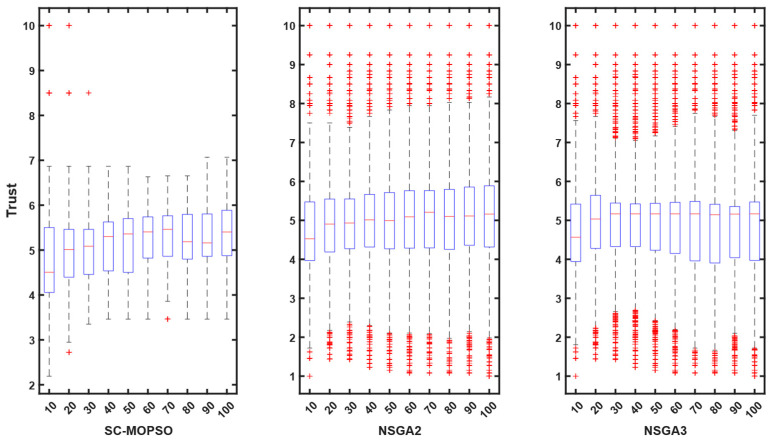
Boxplot representation of trust starting from first generation until the end. The + sign indicates the outliers.

**Figure 8 sensors-23-05526-f008:**
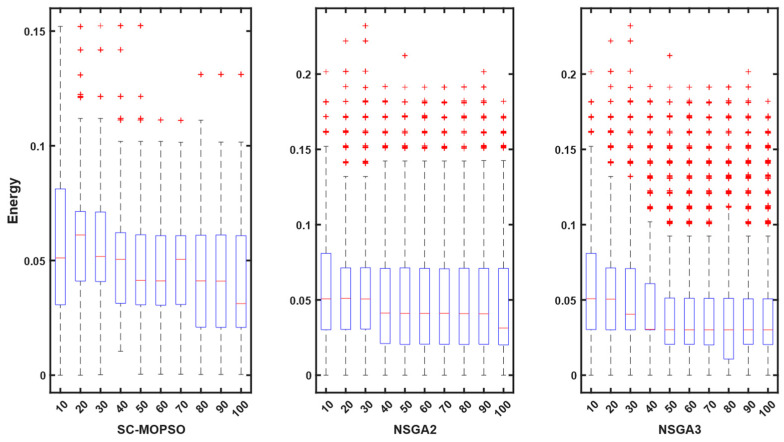
Boxplot representation of energy starting from first generation until the end. The + sign indicates the outliers.

**Figure 9 sensors-23-05526-f009:**
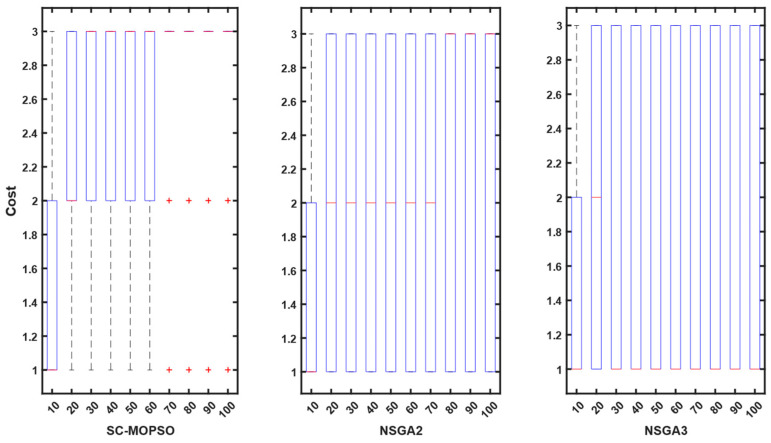
Boxplot representation of cost starting from first generation until the end. The + sign indicates the outliers.

**Figure 10 sensors-23-05526-f010:**
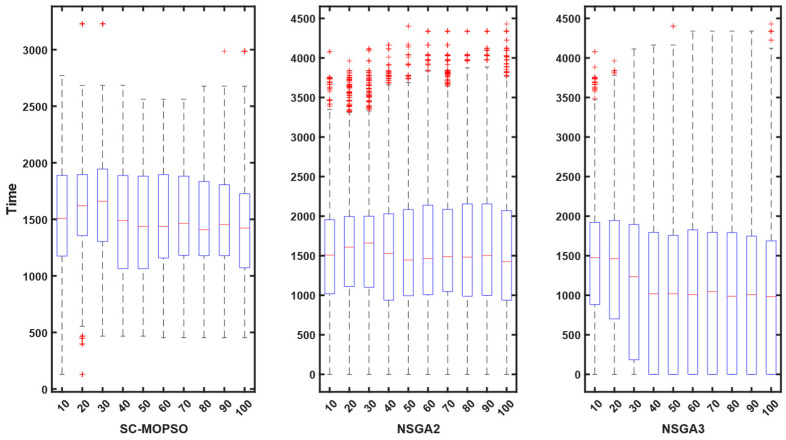
Boxplot representation of time cost starting from first generation until the end. The + sign indicates the outliers.

**Figure 11 sensors-23-05526-f011:**
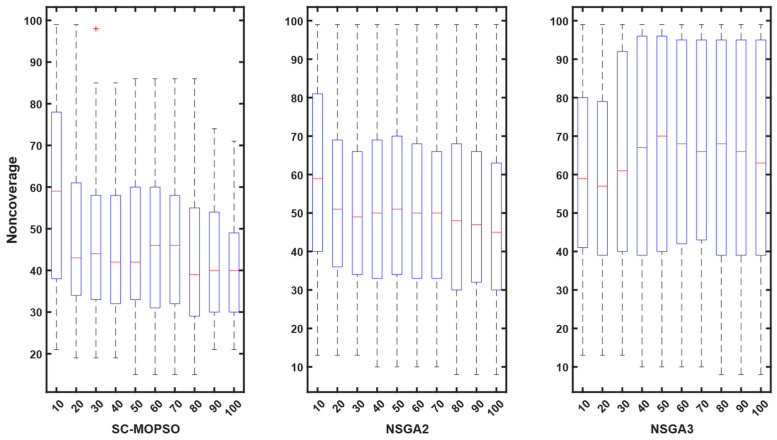
Boxplot representation of the noncoverage from the first generation until the end. The + sign indicates the outliers.

**Figure 12 sensors-23-05526-f012:**
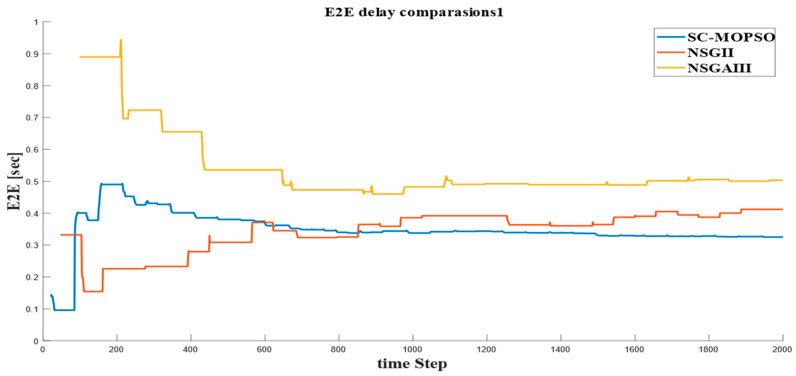
Time series of E2E generated from our developed algorithm and the benchmarks.

**Figure 13 sensors-23-05526-f013:**
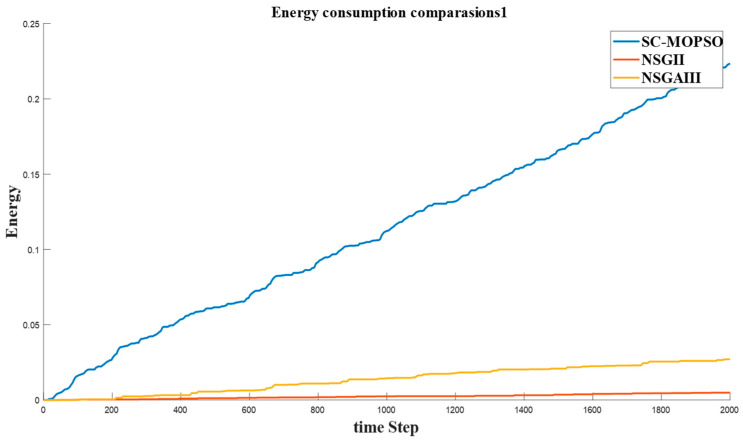
Time series of energy generated from our developed algorithm and the benchmarks.

**Figure 14 sensors-23-05526-f014:**
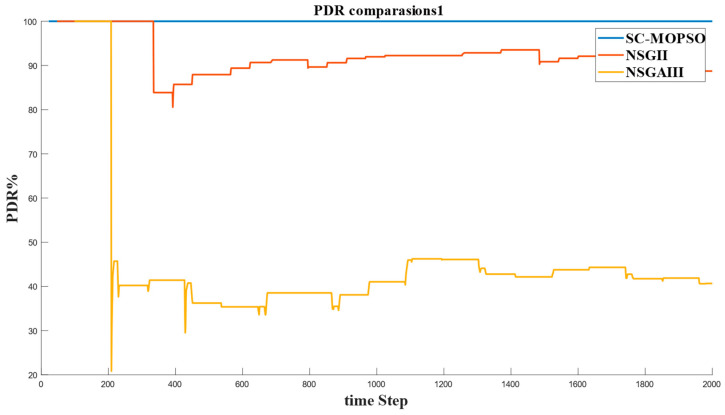
Time series of PDR generated from our developed algorithm and the benchmarks.

**Figure 15 sensors-23-05526-f015:**
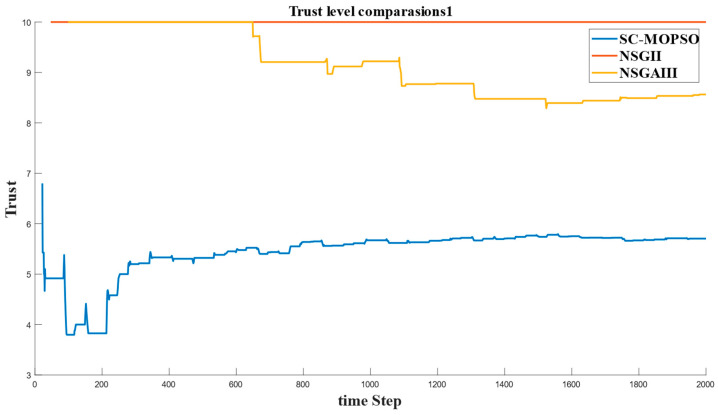
Time series of trust generated from our developed algorithm and the benchmarks.

**Table 1 sensors-23-05526-t001:** Overview of existing MCDM-based approaches in WSNs.

Article	Application	MCDM	Optimization	Domain
[36]	Clustering	TOPSIS	Greedy	WSNs
[37]	Routing	SEAT-DSR	-	WSNs
[39]	Routing	AHP and TOPSIS	-	WSNs
[40]	Sensor selection	-	-	WSNs
[41]	Path planning	-	multiobjective evolutionary algorithms (MOEAs)	WSNs

**Table 2 sensors-23-05526-t002:** An overview of mathematical symbols used in the article and their meaning.

Symbol	Meaning
FAM	Matrix of fitness adjacencies
$N_{C}$	Length of solutions inside the class
RAM	Reduced adjacency matrix
${ram}_{ij}$	Fitness value of class j concerning objective i
WCT	Weak classes threshold
SCT	Strong class threshold
Nms	Number of mobile sinks
$N_{Rv}^{i}$	Number of rendezvous points for mobile sink i
${NH}^{i,j}$	Number of hops at mobile sink i at rendezvous point j
$T_{jk}^{i}$	Trust of sensor that sending data to mobile sink i at rv j
$d_{i}$	Distance travelled by mobile sink i
$N^{i,j}$	Number of sensors at the rendezvous point j for mobile sink i
$E_{i,j,k}$	Energy consumption at sensor k, rendezvous point j, and mobile sink i
${RP}_{j}$	Rendezvous point

**Table 3 sensors-23-05526-t003:** The weights that are given to the criterion.

Criterion	Weight	Value
C(i)	Very Less important	1
	Less important	2
	important	3
	High important	4
	very high importance	5

**Table 4 sensors-23-05526-t004:** Pareto front table.

No	Alternative	C1	C2	C3	C4
Solution x(i)	x(i)	$f_{1}(x\left(i\right))$	$f_{2}(x\left(i\right))$	$f_{3}(x\left(i\right))$	$f_{4}(x\left(i\right))$

**Table 5 sensors-23-05526-t005:** Ranking of solutions provided in the Pareto front table.

No.	Alternative	C1	C2	C3	C4	Ranking
	Weight	$w_{1}$	$w_{2}$	$w_{3}$	$w_{4}$	
Solution x(i)	x(i)	$f_{1}(x\left(i\right))$	$f_{2}(x\left(i\right))$	$f_{3}(x\left(i\right))$	$f_{4}(x\left(i\right))$	$V_{i}=\sum_{j=1}^{n}w_{j}f_{j}(x\left(i\right))$

**Table 6 sensors-23-05526-t006:** Two candidate solutions and their corresponding metameric variables.

Solution ${sol}_{i}=\{{Nms}^{i},N_{Rv}^{i,j},{NH}^{i,j,k},\{\left(x_{i,j,},y\right)\}\}$	Mobile Sink ms(i,j)	Index of RV RV(i,j,k)	POS [x,y]	Index of nHops nh(i,j,k)	NHops
${sol}_{1}=[2,3,2,\left[150\;400\right],\left[200\;144\right],$ $\left[160\;350\right],\left[549\;988\right],\left[234\;633\right],2,3,2,1,3]$	ms(1,1)	RV(1,1,1)	[150 400]	nh(1,1,1)	2
		RV(1,1,2)	[200 144]	nh(1,1,2)	3
		RV(1,1,3)	[160 350]	nh(1,1,3)	2
	ms(1,2)	RV(1,2,1)	[549 988]	nh(1,2,1)	1
		RV(1,2,2)	[234 633]	nh(1,2,2)	3
${sol}_{2}=[3,2,1,3,\left[237,645\right],\left[455,744\right],$ $\left[753,433\right],\left[543,865\right],\left[236,765\right],[653,778]]$	ms(2,1)	RV(2,1,1)	[237 645]	nh(2,1,1)	3
		RV(2,1,2)	[455 744]	nh(2,1,2)	2
	ms(2,2)	RV(2,2,1)	[753 433]	nh(2,2,1)	1
	ms(2,3)	RV(2,3,1)	[543 865]	nh(2,3,1)	2
		RV(2,3,2)	[236 765]	nh(2,3,2)	1
		RV(2,3,3)	[653 778]	nh(2,3,3)	2

**Table 7 sensors-23-05526-t007:** Parameters values of SC-MOPSO and the experimental results.

Parameters	SC-MOPSO	NSGA II	NSGA III
Lower boundary positions	X = 0, y = 0	X = 0, y = 0	X = 0, y = 0
Coverage radius	100	100	100
Higher boundary positions	X = 1000, y = 1000	X = 1000, y = 1000	X = 1000, y = 1000
Lower boundary dimensions	[1,2,2]	[1,2,2]	[1,2,2]
Higher boundary dimensions	[3,4,6]	[3,4,6]	[3,4,6]
Velocity min	[0,0,0]	N/A	N/A
Velocity max	[200,200,2]	N/A	N/A
Number of iterations	100	100	100
Repository size	200	N/A	N/A
Percentage mutation	1	N/A	N/A
Mutated ration	0.5	0.5	N/A
Number of grids	7	N/A	N/A
Alpha	0.1	N/A	N/A
Weight	0.5	N/A	N/A
Scale	0.1	0.1	0.1
Shrink	0.5	0.5	0.5
Fraction	N/A	0.5	0.5
Min No. of particles	3	N/A	N/A
Coverage radius	100	100	100

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
