# Peer review of "Variable-Length Multiobjective Social Class Optimization for Trust-Aware Data Gathering in Wireless Sensor Networks"

_sensors, 2023, doi:10.3390/s23125526_

Round 1
Reviewer 1 Report
Dear Authors of the manuscript
"Variable Length Multiobjective Social Class Optimization for Trust Aware Data Gathering in Wireless Sensor Network"
Summary: This work proposed a modified SC-MOPSO for WSN application where the contributions of this work are
- Add trust-aware data gathering to SC-MOPSO and use SAW for MCDM.
- Evaluation in various matrixes.
The reviewer found the issues as follows.
- Abbrivation presention is not proper. For example, SC-MOPSO first displayed in 57, but no full name is present. The second example is E2E-delay which never explains the full name. SEEGT also is never defined. Please check more in the attached file.
- English problem. For example, Line 108, the -> The. Line 137 “In the work [27]. actuality” no need “.”. Please check more in the attached file.
- The major issue is about explaining the proposed idea. The author uses only Algorithms to explain the process of the proposed concept. It will help a lot if we can have a diagram or flowchart to explain the proposed idea and also the traditional one.
- Authors denote the proposed algorithm as “SC-MOPSO-DG,” but it never uses in the experimental result. We found only SC-MOPSO in the result. Please explain.
- In the result, NSGA II and NSGA III are used for comparison, which these two algorithms never explained before.
- Figures 7-10 need to be mentioned in the text and need to be explained.
- No discussion is found in this manuscript.
- References number should be a number without bracket.
- Table 1 in Appendix A, how do the articles have been sorted? One suggestion is sorting by the reference number.
- What is the difference between SC-MOPSO-DG and SC-MOPSO? Can we compare the result of these two algorithms?

Author Response
Dear Reviewer,
Thank you very much for your valuable input on our manuscript. We really appreciate every single feedback shared with us for the improvement of the article. This exercise has provided us an opportunity to enhance our research insides.
Following are the list of review outcome that has been addressed to the best of our understanding. Appreciate further assistance if there is further improvement required.
Thank you.
Regards,
Kalaivani Chellappan

Reviewer 2 Report
The manuscript titled “Variable Length Multi-objective Social Class Optimization for Trust Aware Data Gathering in Wireless Sensor Network” propped a novel method based on “trust”. The work seems sufficient and supported by results which were discussed nicely.
The paper is nicely presented; however, the following comments/suggestions are included for revision.
- What is trust and how you have quantified the value of trust is not mentioned in the paper?
- What is the threshold value used of the trust for which you will consider the node/sink trustworthy?
- The authors claimed that the decision maker decides the weights. On what factor and how the weights are assigned? What are the features considered for assigning the weights? Have you done a feature importance analysis?
- The proposed algorithm is compared with the benchmarks with only 100 iterations. What will be the performance if the number of iterations is >100? Find the impact.
- Make a flow diagram for the fault detection for ready reference of the readers.
- Every work has some limitations and assumptions. Try to document the same for your proposed model concisely.
- Check the axes and title of the graphs and tables carefully and correct them if required.
- In table 5, X and y are in uppercase and lowercase, why it is so?
- The unit of coverage radius is not specified in table 5.
- Derive the fitness function so that the readers may follow your paper.
Author Response

(The authors gave the same response as above.)

Reviewer 3 Report
1) Please review the manuscript for grammatical and spelling errors. There are a few just in the introduction like in the first paragraph of the introduction is it “stationary slinks” (line 34) or “stationary sinks”, “hopes” (line 39).
2) One line 45 “Multiobjective optimization involves problems [10] that have more than one objective with conflicting nature [11].” Please add a few sentences after this to explain what those conflicting objectives are and why are they conflicting.
3) Please include a few real-life applications to show the importance of trust-aware data gathering in wireless sensor networks.
4) The algorithm “SC-MOPSO-DG” is extremely hard to follow. Please include working examples of each algorithm so that readers can comprehend these algorithms.
5) Please give the readers insights into the main steps of your proposed algorithm.
6) In “Experimental Works and Results” please specify the platform on which these experiments were conducted.
7) “NSGA II” and “NSGA III” terms are referred to in the text but never defined and explained. Although “NSGA-II: Non-dominated Sorting Genetic Algorithm” is a well know optimization algorithm it is still important to define these terms and explain them briefly. Please briefly describe how “NSGA II” and “NSGA III work in the “Experimental Works and Results” section.
8) Please define or at least explain terms such as “set coverage”, and “Hyper Volume”. The readers may associate meaning to these terms based on their experience and knowledge.
9) There are no reasons given for the superior performance of the proposed approach compared to existing ones. Please discuss why the proposed approach is better.
Author Response

(The authors gave the same response as above.)

Reviewer 4 Report
The authors proposes a modified social class multiobjective particle swarm optimization (SC-MOPSO). The modified SC-MOPSO is featured with application-dependent operators which were named inter-class operators, and thoroughly explained in the study. To validate this proposal the authors compare their method with 2 others in a network simulator.
In a general overview the paper is well written and its well structured. A few phrases that could be better complemented or re-written:
Line 66 - missing the structure of the paper.
Line 69 - could not understand what authors are trying to say
Lines 482-483 - missing the subsection numbers
Line 488 - typo "Pparameters"
Main concern of the paper is in the results section. The validation against two other methods is of the utmost importance and the authors present many of the results they obtained. The problem is that it lacked its discussion. From Figure 6 to Figure 13, the authors did not discuss what was observable in these Figures and if SC-MOPSO really outperforms the others.
Lastly, the conclusions seem rather short considering the amount of results provided.
Author Response

(The authors gave the same response as above.)

Round 2
Reviewer 1 Report
Thank you for your revision. However, as in the following comment, the revised version still has many things to correct.
1. The revised document is difficult to check. If you use Microsoft word, please use the track change feature. If you use latex, please use latexdiff
https://www.overleaf.com/learn/latex/Articles/Using_Latexdiff_For_Marking_Changes_To_Tex_Documents
2. English still has a lot of mistakes, especially in the punctuation
3. In the revised manuscript, please label the highlight. So we can know what correction has been made.
Please refer to the previous review and make the proper correction.
Best Regards
Author Response
Reviewer 1
Dear Reviewer,
Thank you very much for your valuable input on our manuscript. We really appreciate every single feedback shared with us for the improvement of the article. This exercise has provided us an opportunity to enhance our research insides.
We have completed another round of language editing on punctuation and spelling errors as advised. Following has been addressed accordingly:
- The revised document is difficult to check. If you use Microsoft word, please use the track change feature. If you use latex, please use latex;
Track change has been added as requested.
- English still has a lot of mistakes, especially in the punctuation
Another round of proof reading has been attempted as advised.
- In the revised manuscript, please label the highlight. So we can know what correction has been made.
Track change has been added as requested.
- Please refer to the previous review and make the proper correction.
Has addressed all the requested change in our best capacity possible. Please advice if there is further correction required.
Thank you.
Regards,
Kalaivani Chellappan

Reviewer 3 Report
The authors have addressed most of the issues. I have no further comments or suggestions.
Author Response
Reviewer 3
Dear Reviewer,
Thank you very much for your valuable input on our manuscript. We really appreciate every single feedback shared with us for the improvement of the article. This exercise has provided us an opportunity to enhance our research insides.
We have completed another round of language editing on punctuation and spelling errors as advised.
Thank you.
Regards,
Kalaivani Chellappan

Reviewer 4 Report
The authors corrected and revised the enumerated points.
Author Response
Reviewer 4
Dear Reviewer,
Thank you very much for your valuable input on our manuscript. We really appreciate every single feedback shared with us for the improvement of the article. This exercise has provided us an opportunity to enhance our research insides.
We have completed another round of language editing on punctuation and spelling errors as advised.
Thank you.
Regards,
Kalaivani Chellappan
